# Epidermal Fatty Acid-Binding Protein 5 (FABP5) Involvement in Alpha-Synuclein-Induced Mitochondrial Injury under Oxidative Stress

**DOI:** 10.3390/biomedicines9020110

**Published:** 2021-01-22

**Authors:** Yifei Wang, Yasuharu Shinoda, An Cheng, Ichiro Kawahata, Kohji Fukunaga

**Affiliations:** Department of Pharmacology, Graduate School of Pharmaceutical Sciences, Tohoku University, 6–3 Aramaki-Aoba, Aoba-ku, Sendai 980-8578, Japan; yifei.wang.q4@dc.tohoku.ac.jp (Y.W.); yshinoda@tohoku.ac.jp (Y.S.); cheng.an.q6@dc.tohoku.ac.jp (A.C.); kawahata@tohoku.ac.jp (I.K.)

**Keywords:** α-Synuclein, FABP5, aggregation, mitochondria, Parkinson’s disease

## Abstract

The accumulation of α-synuclein (αSyn) has been implicated as a causal factor in the pathogenesis of Parkinson’s disease (PD). There is growing evidence that supports mitochondrial dysfunction as a potential primary cause of dopaminergic neuronal death in PD. Here, we focused on reciprocal interactions between αSyn aggregation and mitochondrial injury induced by oxidative stress. We further investigated whether epidermal fatty acid-binding protein 5 (FABP5) is related to αSyn oligomerization/aggregation and subsequent disturbances in mitochondrial function in neuronal cells. In the presence of rotenone, a mitochondrial respiratory chain complex I inhibitor, co-overexpression of FABP5 with αSyn significantly decreased the viability of Neuro-2A cells compared to that of αSyn alone. Under these conditions, FABP5 co-localized with αSyn in the mitochondria, thereby reducing mitochondrial membrane potential. Furthermore, we confirmed that pharmacological inhibition of FABP5 by its ligand prevented αSyn accumulation in mitochondria, which led to cell death rescue. These results suggested that FABP5 is crucial for mitochondrial dysfunction related to αSyn oligomerization/aggregation in the mitochondria induced by oxidative stress in neurons.

## 1. Introduction

Parkinson’s disease (PD) is the second-most common neurodegenerative disease, and is caused by the loss of dopaminergic neurons in the substantia nigra pars compacta (SNpc) [1]. The neuropathological hallmark of PD is the accumulation of intracellular protein inclusions composed primarily of α-synuclein (αSyn), which are termed Lewy bodies [2,3]. Moreover, αSyn aggregation has been widely reported to be a harbinger of subsequent pathology that ultimately leads to neurodegeneration [4]. αSyn is expressed pre-synaptically as a native unfolded protein that can form oligomers, protofibrils, or amyloid fibrils [5,6]. Among the different species of αSyn, pre-fibrillar oligomers are considered to be toxic to neurons [7]. The development of treatments aimed at reducing αSyn synthesis, secretion, aggregation, and increasing the clearance of pathogenic formations has become the most promising means to overcome Parkinson’s disease [8,9].

Several lines of evidence support mitochondrial dysfunction as a primary pathogenic mechanism in PD. The most convincing evidence first appeared from accidental human exposure to 1-methyl-4-phenyl-1,2,3,6-tetrahydrodropyridine, a metabolite that induces Parkinsonian syndrome by inhibiting mitochondrial respiratory complex I activity [10,11]. The genetic causes of PD, including *PINK1*, *parkin*, and *DJ-1*, particularly highlight the importance of mitochondrial dysfunction in PD pathology [12]. Accumulating studies have demonstrated that αSyn itself is involved in mitochondrial injury in PD [13,14,15]. In postmortem brains of PD patients, αSyn is especially localized to mitochondrial outer membranes in nigrostriatal neurons, which was replicated in rodents treated with rotenone, a respiratory chain complex I inhibitor [14]. Treatment of αSyn oligomers, but not monomers, disturbed mitochondrial maintenance, leading to fission, reduction in mitochondrial membrane potential, and upregulation of reactive oxidative stress in neuroblastoma cells. Preformed fibril treatment also elicited the translocation of phosphorylated αSyn aggregates to mitochondria, resulting in increased oxidative stress and mitochondrial dysfunction in mouse primary hippocampal neurons [13]. These results indicate the bilateral links that oligomeric αSyn localization in mitochondria induces reactive oxidative species production/mitochondrial dysfunction, and vice versa.

Fatty acid-binding proteins (FABPs) are a set of 14–15 kDa cytoplasmic proteins that are traditionally considered as lipid chaperones that coordinate lipid responses inside cells. FABPs perform pleiotropic functions to maintain healthy tissue homeostasis, and participate in the pathogenesis of diseases [16]. To date, at least 10 genes encoding FABPs (*FABP1-9* and *FABP12*) have been identified in the human genome [17,18]. FABP5 localizes to the cytosol, mitochondria, and nucleus, suggesting a role for fatty acid metabolism and nuclear transcriptional regulation [19,20]. The central nervous system contains three types of FABP, that is, epidermal FABP (E-FABP, FABP5), heart FABP (H-FABP, FABP3), and brain FABP (B-FABP, FABP7) [21]. FABP5 is expressed in neurons and glial cells in broad areas of the brain, including the cortex, hippocampus, and caudate putamen [22]. Moreover, it has been reported that *FABP5* mRNA expression levels, as well as αSyn and tyrosine hydroxylase gene expression levels, are enriched in dopaminergic neurons of the SNpc, and decrease after 6-hydroxydopamine injection in rats [23]. However, little is known regarding whether FABP5 is involved in αSyn toxicity in dopaminergic neurons, or in the pathogenesis of PD.

To investigate whether FABP5 is related to αSyn oligomerization and mitochondrial impairment, we transfected FABP5 with αSyn into neuroblastoma Neuro-2A cells, and evaluated cell viability, αSyn oligomerization, cellular localization, and mitochondrial membrane potential. Here, we report that FABP5 co-expression with αSyn reduced cell viability, upregulated αSyn oligomerization, and aggregation under oxidative stress. FABP5 and αSyn co-localized in mitochondria in the presence of rotenone, which was abolished by a ligand with high affinity for FABP5.

## 2. Materials and Methods

### 2.1. Materials

Reagents and antibodies were obtained from the following sources: anti-αSyn antibody (1:1000; 4B12, GeneTex, Irvine, CA, USA); anti-FABP5 antibody (1:1000; AF3077, R&D Systems, Minneapolis, MN, USA); anti-β-tubulin antibody (1:3000; T0198, Sigma-Aldrich, St Louis, MO, USA); anti-VDAC antibody (1:1000; 4866S, Cell Signaling Technology, Danvers, MA, USA); anti-rabbit or mouse IgG antibody conjugated with horseradish peroxidase (1:5000; Southern Biotech, Birmingham, AL, USA); anti-goat IgG antibody conjugated with peroxidase (1:5000; Rockland Immunochemical, Limerick, PA, USA); Alexa 594-labeled anti-mouse IgG, Alexa 488-labeled anti-Goat IgG and Alexa 594-labeled anti-rabbit IgG antibody (1:500; Invitrogen, Waltham, MA, USA); Biotin-SP anti-Goat IgG antibody (1:500; Jackson ImmunoResearch, West Grove, PA, USA); and AMCA streptavidin (1:500; Jackson ImmunoResearch). These materials were used according to the manufacturers’ datasheets. Rotenone was purchased from MP Biomedicals (150154; Santa Ana, CA, USA). All other reagents were obtained from FUJIFILM Wako Pure Chemicals (Osaka, Japan) unless otherwise noted. The FABP5 ligand (ligand 7) was described in a previous report [24].

### 2.2. Cell Culture and Transfection

Neuro-2A cells were obtained from the Human Science Research Resources Bank (IFO50081) (Osaka, Japan). Cells were maintained in Dulbecco’s modified Eagle medium (DMEM) supplemented with 10% fetal bovine serum (FBS) and 100 U/mL penicillin-streptomycin at 37 °C in a humidified atmosphere of 5% CO_2_. For stimulation, rotenone and FABP ligands were prepared in dimethyl sulfoxide (DMSO). Mouse FABP5 plasmid was produced as described previously [24], and inserted into the multiple cloning site of plasmid pcDNA3.1 (Invitrogen). Human αSyn plasmid encoding the protein with myc and 6×His tag at the C-terminus was purchased from Abgent (San Diego, CA, USA). A non-coding plasmid (pcDNA3.1) was used for a mock group. Transfection was carried out using Lipofectamine LTX and Plus Reagent (Invitrogen) by incubating for 6 h. The medium was then changed to DMEM supplemented with 5% FBS to maintain cell viability. The next day, cells were treated with rotenone, ligand 7, or DMSO for 48 h at a dilution of 1:1000, and then used for the following experiments.

### 2.3. Evaluation of Cell Viability

Cells were cultured in 96-well microplates, and 1:10 dilution of CCK-8 solution was added after transfection and drug treatment (Cell Counting Kit-8; Dojindo, Kumamoto, Japan). Plates were incubated for 2 h at 37 °C, and the optical absorbance at 450 nm was measured using a microplate reader (Flex Station 3, Molecular Devices, San Jose, CA, USA).

### 2.4. Protein Extraction and Immunoblotting Assay

Cells were cultured in a 35 mm dish, collected by scraping, and homogenized in 80 µL buffer containing 0.5% Triton X-100, 50 mM Tris-HCl, pH 7.4, 4 mM EGTA, 10 mM EDTA, 1 mM Na_3_VO_4_, 40 mM Na_4_P_2_O_7_·10H_2_O, 50 mM NaF, 0.15 M NaCl, and protease inhibitor cocktail (50 µg/mL leupeptin, 25 µg/mL pepstatin A, 50 µg/mL trypsin inhibitor, 100 nM calyculin A, and 1 mM dithiothreitol). Then, lysates were centrifuged at 15,000 rpm (10,000 × *g*) for 10 min at 4 °C, the supernatants (Triton-soluble fraction) collected, and protein concentrations were measured using the Bradford assay. Insoluble materials were washed three times and centrifuged at 15,000 rpm for 3 min at 4 °C, and the supernatant was discarded. The pellets were homogenized in 2% SDS solution (SDS-soluble fraction). In order to ensure equivalent protein loading, the volume of the 2% SDS solution was proportional to the concentration of Triton-soluble protein. Mixtures with Laemmli’s sample buffer without β-mercaptoethanol were subjected to heating before immunoblotting to evaluate αSyn oligomerization.

For immunoblotting assays, equal amounts of protein were electrophoresed on ready-made gels (414886; Cosmo Bio, Tokyo, Japan) and transferred onto PVDF membranes (Millipore, Billerica, MA, USA). After blocking with 5% skim milk in Tris-buffered saline with Tween 20, the membrane was incubated on a horizontal shaker overnight with primary antibody at 4 °C, followed by probing with secondary antibody for 2 h at 20 °C. Signals were detected by chemiluminescence using an imaging analyzer (LAS4000 Mini, Fuji Film, Tokyo, Japan). Densitometry analysis was performed using Multi Gauge Software (FUJIFILM). Briefly, we framed each target band and also framed the part without bands as the background to get their densities. The difference between the two is the intensity of the target band and is standardized with the internal reference protein (β-tubulin). Finally, the values were normalized with the mock group before statistical analysis.

### 2.5. Immunofluorescent Staining

Neuro-2A cells were cultured on glass slips in 12-well plates (Corning, Glendale, AZ, USA) and fixed in 4% paraformaldehyde for 10 min at room temperature. For mitochondrial staining, cells were incubated with 0.1mM MitoTracker Red CMXRos dye (M7512; Invitrogen) in DMEM without FBS for 30 min at 37 °C before fixation. Cells were washed in phosphate-buffered saline (PBS) and permeabilized with 0.1% Triton X-100/PBS for 10 min, then blocked in 1% BSA/PBS at room temperature for 30 min. Cells were incubated with primary antibodies at 4 °C overnight. After washing in 1% BSA/PBS for 5 min three times, cells were incubated with secondary antibodies for 2 h at 4 °C. Cells were washed in 1% BSA/PBS for 5 min three times, followed by staining with 4′,6-diamidino-2-phenylindole (DAPI) (Thermo Fisher Scientific, Waltham, MA, USA). Glass slips were mounted with Vectashield (Vector Laboratories, Burlingame, CA, USA). Immunofluorescent images were captured with a confocal laser scanning microscope (TCS SP8, Leica Microsystems, Wetzlar, Germany). The visible aggregates (diameter 0.5–2 µm approximately) were quantitated.

### 2.6. Measurement of Mitochondrial Membrane Potential

Cells were cultured in 96-well plates (Corning) or glass-bottom dishes (Matsunami Glass Ind, Osaka, Japan). Mitochondrial activity was visualized using JC-1 MitoMP Detection Kit (Dojindo) staining, according to the manufacturer’s instructions. JC-1 is a cationic dye that exhibits membrane potential-dependent accumulation in mitochondria, which is indicated by a fluorescence emission shift from green (~530 nm) to red (~590 nm). The fluorescence intensity was measured with Flex Station 3, or using ImageJ software following acquisition of images by confocal microscopy [25]. The ratio of red/green fluorescence intensity was calculated and converted into a value relative to the control group for the identification of mitochondrial membrane potential.

### 2.7. Mitochondrial Purification

Mitochondria and cytoplasm were isolated from Neuro-2A cells as described previously [26] with slight modifications. Briefly, cells were suspended in mitochondrial isolation buffer containing 250 mM sucrose, 1 mM dithiothreitol, 10 mM KCl, 1 mM EDTA, 1.5 mM MgCl_2_, protease inhibitors, and 20 mM Tris-HCl, pH 7.4, and then homogenized with a glass homogenizer using approximately 30 strokes, on ice. The homogenate was centrifuged twice at 800× *g* for 10 min. The supernatant was further centrifuged at 15,000× *g* for 10 min. After that, supernatants were collected as cytosolic fractions (without mitochondria). Next, the mitochondrial pellets were immediately washed three times in mitochondrial isolation buffer. Subsequently, the pellets were homogenized with Triton X-100 buffer as described above, and supernatants were collected as mitochondrial fractions. All steps were carried out at 4 °C. Protein concentrations were determined by using Bradford assays. All fractions were boiled at 100 °C for 3 min with six × Laemmli sample buffer containing β-mercaptoethanol. The quantity of cytosolic fractions and mitochondria were, respectively, estimated by quantification of β-tubulin and VDAC using an immunoblotting assay.

### 2.8. Statistical Analysis

Values are presented as means ± standard error of the mean (SEM), and were evaluated with one- or two-way analysis of variance (ANOVA) followed by Bonferroni’s multiple comparisons test using GraphPad Prism 7.04 (GraphPad Software, San Diego, CA, USA). *p* < 0.05 was considered statistically significant.

## 3. Results

### 3.1. Co-Overexpression of FABP5 and α-Synuclein Increases Neurotoxicity in the Presence of Rotenone

To address the role of FABP5 in the toxicity of αSyn and oxidative stress, Neuro-2A cells were transfected with plasmids and then stimulated with 0.01–10 μM rotenone (Figure 1). Immunoblotting analysis showed moderate levels of endogenous FABP5 protein and higher levels of FABP5 and αSyn in transfected cells (Figure 1A). There is no FABP3 expression and a very little FABP7 expression in Neuro-2A cells (Appendix A). CCK-8 assays revealed that rotenone at concentrations > 1 μM significantly decreased cell viability in mock- and FABP5-transfected cells (Figure 1B). Similar toxicity of oxidative stress was demonstrated in αSyn-transfected cells, whereas above 0.5 μM, rotenone treatment further decreased cell viability in αSyn/FABP5 (αS/F5)-expressing cells (Figure 1C). This effect of rotenone was evident when treated at a moderate concentration (0.5 μM), which failed to decrease cell survival in αSyn-transfected cells. These results suggested that FABP5 enhances oxidative stress toxicity in the presence of αSyn.

### 3.2. Rotenone Induces α-Synuclein and FABP5 Oligomerization and Aggregation

We next asked whether the toxicity of rotenone is associated with αSyn oligomerization or aggregation. We separated cell lysates into Triton-soluble and SDS-soluble fractions, as evaluated by immunoblotting under non-denatured conditions (Figure 2). αSyn and FABP5 monomers remained in the Triton-soluble fractions (Figure 2A,C). Meanwhile, the levels of αSyn oligomers (70–140 kDa) and aggregates (>210 kDa) significantly increased following 0.1 μM rotenone treatment (Figure 2B,F,G). αSyn dimer/trimer levels were also upregulated, but this difference was not statistically significant (Figure 2E). Furthermore, immunoblotting analysis with αSyn antibody and re-blotting with FABP5 antibody showed that there were also oligomers and aggregates at the same molecular weights as αSyn (Figure 2D). Quantitative analysis revealed upregulation of dimer/trimer (30–55 kDa), oligomers (70–140 kDa), and aggregates (>210 kDa) in rotenone-treated cells, but there was a statistically significant difference only in term of aggregate levels (Figure 2E–G). The negative control of immunoblotting was shown in Appendix A, and there is no target band detected. These results suggested that FABP5 cooperates with αSyn oligomerization and aggregation under oxidative stress.

### 3.3. FABP5 Co-Localizes with αSyn Aggregates Induced by Oxidative Stress

Next, cells were subjected to immunofluorescent staining to analyze the subcellular localization of αSyn and FABP5 (Figure 3). The signals detected for both αSyn and FABP5 showed a diffused cellular pattern of localization (Figure 3A). However, when treated with 0.1 or 0.5 μM rotenone, immunolabeling revealed visible significant αSyn accumulation in FABP5/αSyn co-overexpressing cells (Figure 3B, arrow). FABP5 immunoreactivity co-localized with αSyn aggregates in rotenone-treated cells (Figure 3B), suggesting that FABP5 forms complexes with αSyn aggregates. Taken together, these results suggested that rotenone promotes FABP5 and αSyn interactions, thereby causing αSyn oligomer and aggregate formation.

### 3.4. FABP5 is Involved in Mitochondrial Membrane Potential Reduction by Rotenone

We further employed the JC-1 assay to detect mitochondrial membrane potentials (Figure 4). Reductions in mitochondrial membrane potentials are indicated by a decrease in the ratio of red to green fluorescence of the JC-1 dye. First, we evaluated the effects of transfection on cells and found that there were no obvious effects (Figure 4A,C). Meanwhile, treatment with rotenone at 0.1 and 0.5 μM significantly decreased JC-1 red fluorescence and the ratio (red/green), especially in *αSyn/F5* cells (Figure 4B,D). The ratio at 0.1 μM rotenone was not altered in αSyn-expressing cells, but was significantly reduced in *αSyn/F5* cells (Figure 4D).

### 3.5. FABP5 Ligand Rescues Rotenone-Induced Cell Death and Impedes αSyn Oligomerization and Aggregation

Next, we evaluated whether FABP5 ligand can abolish oxidative stress toxicity and αSyn oligomerization. BMS309403 was identified as a small-molecule FABP4-selective inhibitor, with a particularly high affinity for FABP4 [27]. Previously, we developed several BMS309403 derivatives and reported their affinity for GST-FABP5 [24,28]. Based on these studies, we used ligand 7, which has high affinity for FABP5, with a dissociation constant (*Kd*) of 199 ± 23 nM, which is almost identical to that of the intrinsic fatty acid, arachidonic acid (AA) at 228 ± 43 nM (Figure 5A,B) [24]. Treatment of ligand 7 increased cell viability after 0.5 μM rotenone treatment, and there was a statistically significant difference at 1 and 3 μM ligand 7 (Figure 5C). The reduction in the JC-1 ratio (red/green) by rotenone was also significantly improved by treatment with 1 μM ligand 7 (Figure 5D). We further analyzed oligomerization and aggregation of αSyn and FABP5 in the SDS-soluble fraction, which was apparently suppressed by ligand 7 (Figure 5E,F). At the same time, the monomers of αSyn and FABP5 remained in Triton-soluble fractions, which is consistent with Figure 2A,C and Figure 5G. These results suggested that FABP5 inhibitor (ligand 7) effectively attenuated αSyn oligomerization and toxicity by conserving mitochondrial function.

### 3.6. FABP5/αSyn Aggregates Targeting Mitochondria and Ligand 7 Prevent Abnormal Accumulation

To further evaluate the mechanisms involved in protection, we isolated mitochondria from *αSyn/F5* cells (Figure 6A). Crude mitochondrial and cytosolic fractions were separated and subjected to immunoblotting assays (Figure 6B,C). As a result, FABP5 and αSyn abnormally accumulated in mitochondria in 0.5 μM rotenone-treated cells, which was prevented by 1 μM ligand 7 treatment (Figure 6D,E). Consistent with these immunoblotting results, αSyn and FABP5 were co-localized with the mitochondrial marker MitoTracker in rotenone-treated cells (Figure 6F, arrow), which was blocked by ligand 7 treatment (Figure 6F).

## 4. Discussion

In this study, we demonstrated that FABP5 plays a key role in oligomerization and aggregation of αSyn under oxidative stress in Neuro-2A cells. In the absence of rotenone, both αSyn and FABP5 localized with diffuse patterns in these cells (Figure 3A); however, their aggregates appeared concentration-dependent on rotenone (Figure 3B–D). Rotenone also led to aggregate localization in mitochondria, and decreased mitochondrial membrane potential, which might have resulted in cell death (Figure 7). We also found that FABP5 ligand 7 treatment abolished the toxicity, αSyn aggregation, and mitochondrial impairment in Neuro-2A cells.

The most basic function of FABPs is to participate in the utilization of lipids in foodstuffs, and to mediate fatty acid transport to various organelles, including mitochondria. A recent study indicated that both acute shRNA knockdown of FABP5 and pharmacologic inhibition results in decreased mitochondrial mass and respiration in regulatory T cells [29]. FABP5 induces mitochondrial macropore formation through BAX and voltage-dependent anion channels (VDAC-1), which mediate mitochondrial outer membrane permeabilization and ultimately accelerates mitochondria-induced oligodendrocyte death. However, FABP7 does not localize in mitochondria under mitochondrial damage [30]. In FABP3-overexpressing embryonic cancer cells (P19 cells), lower cellular ATP production was accompanied by a dramatic decrease in mitochondrial membrane potential, and FABP3 overexpression also led to an imbalance in mitochondrial dynamics and excess intracellular reactive oxygen species production [31]. These studies suggest that FABP proteins regulate mitochondrial function by transporting fatty acids and reactive oxygen species production in cells. In this study, however, we failed to confirm the changes to mitochondrial membrane potential by FABP5 overexpression alone (Figure 4).

Various stimuli, including oxidative stress, can trigger translocation of αSyn to the mitochondria. Transient or stable transfection of αSyn particles exhibit localization in the mitochondria of neuroblastoma SHSY or HEK293 cells [32,33]. Expression of αSyn induces cytochrome C release from mitochondria, or vulnerability to rotenone in these cultured cells. Cellular acidification is also reported to enhance mitochondrial αSyn accumulation [34]. In this study, we could not confirm αSyn localization to mitochondria, or upregulated toxicity of rotenone by overexpression of αSyn alone. Oxidative stress induced by rotenone triggered toxicity and αSyn oligomerization and mitochondrial injury was aggravated by FABP5 co-expressed in Neuro-2A cells (Figure 6).

Aggregation and mitochondrial localization of αSyn and FABP5 may underlie the toxic mechanisms induced by rotenone. Reactive oxygen species produce oxidized aldehydes of polyunsaturated fatty acids such as 4-hydroxy-2-nonenal (4-HNE), 4-hydroxy-2-hexenal, and malondialdehyde [35]. FABP4, another FABP family protein, binds not only with fatty acids, but also with metabolites such as 4-HNE [36]. Fatty acids are usually located in the internal pocket of FABPs; however, 4-HNE can undergo covalent binding with a cysteine residue at amino acid 117. This suggests that strong modifications of FABP protein leads to prolonged regulation of its function. Such modification may underlie mechanisms involving FABP5 localization together with αSyn in mitochondria, because this cysteine residue is also conserved in FABP5, and ligand 7 treatment inhibited aggregation and mitochondrial localization (Figure 6) [24]. However, further studies are required to evaluate the underlying mechanisms whereby FABP5 and αSyn aggregations localize to mitochondria.

Finally, we investigated whether FABP5 ligand treatment prevented its aggregation and localization with αSyn induced by rotenone. FABP5 is highly expressed in the dopaminergic neurons of SNpc [23]; however, little is known regarding the physiological relevance in the neurons or pathological roles of FABP5 in PD. Our study provides evidence that FABP5 protein is a potential therapeutic target in PD therapy.

## Figures and Tables

**Figure 1 biomedicines-09-00110-f001:**
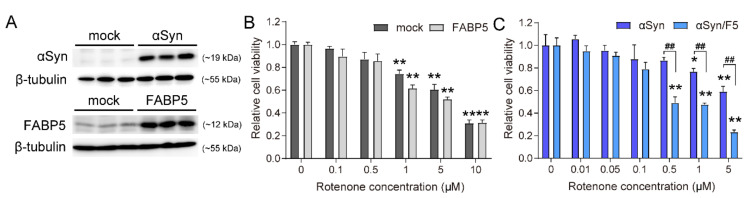
Rotenone induced neurotoxicity, especially with co-overexpression of FABP5 and α-Synuclein. (**A**) Representative images of immunoblots probed with antibodies against αSyn (~19 kDa) and FABP5 (~12 kDa). Blots with anti-β-tubulin antibody showed that a similar amount of protein was loaded. (**B**,**C**) Viability of Neuro-2A cells treated with indicated concentrations of rotenone for 48 h was assessed using CCK-8 assays. The transfection conditions were pcDNA 3.1-transfected cells (mock) and FABP5-transfected cells in (**B**); αSyn-transfected cells and αSyn/FABP5-transfected cells (αS/F5) in (**C**). Results are presented as means ± SEM (n = 4 parallel cell experiments). * *p* < 0.05, ** *p* < 0.01, vs. 0 μM rotenone groups; ## *p* < 0.01 vs. αSyn-transfected cells. CCK-8: cell counting kit-8.

**Figure 2 biomedicines-09-00110-f002:**
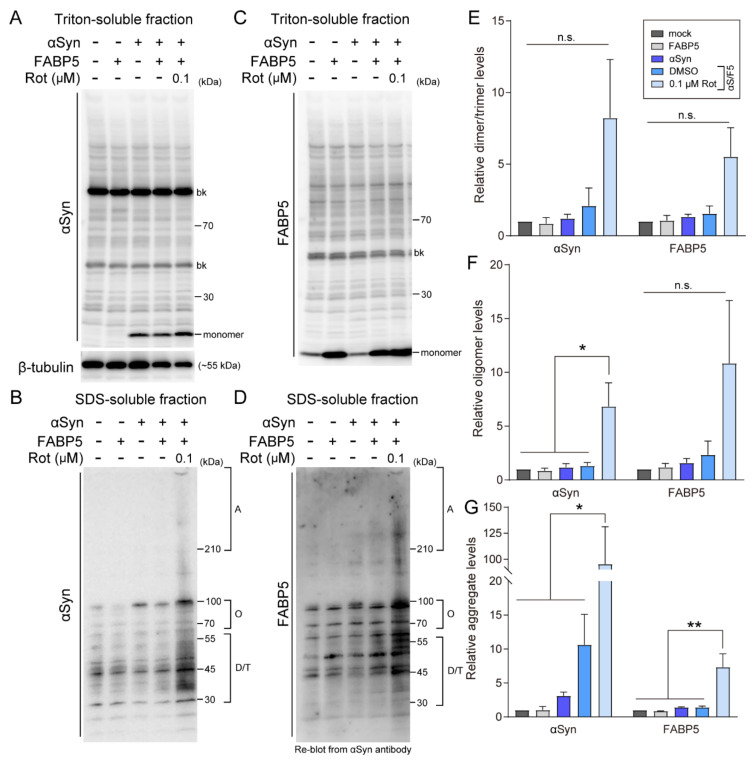
Effects of rotenone on αSyn oligomerization and aggregation in FABP5/αSyn-transfected Neuro-2A cells. (**A**–**D**) Representative images of immunoblots probed with antibodies against αSyn and FABP5 under non-denaturing conditions of Triton-soluble and SDS-soluble fractions. Detection with anti-β-tubulin antibody was used as a protein-loading control. (**E**–**G**) Quantitative analyses of protein contents of different molecular weights, which indicate protein complexes of dimer/trimers, oligomers, and aggregates. Results are presented as means ± SEM (*n* = 3 respective cell experiments). * *p* < 0.05, ** *p* < 0.01, vs. 0.1 μM rotenone-treated αS/F5 cells. n.s., not statistically significant. D/T: dimers/trimers; O: oligomers; A: aggregates. bk besides blots indicate non-specific bands.

**Figure 3 biomedicines-09-00110-f003:**
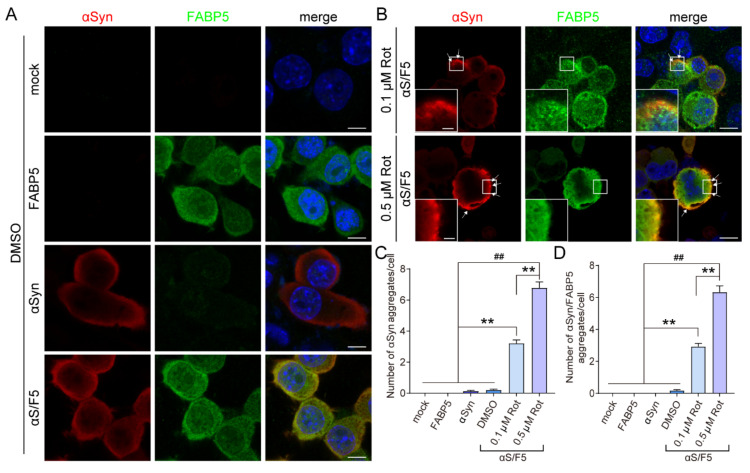
FABP5 co-localized with rotenone-induced α-Synuclein aggregates. (**A**,**B**) Representative images of immunofluorescent staining showing localization of αSyn (red), FABP5 (green), and DAPI (blue) in Neuro-2A cells. (**A**) Cells were transfected with pcDNA 3.1, *FABP5*, *αSyn*, and *αS/F5* respectively, then exposed to DMSO. Scale bars: 10 µm. (**B**) *αS/F5*-transfected cells were exposed to 0.1 or 0.5 μM rotenone. Arrows indicate αSyn aggregates in cells treated with rotenone. Scale bars: 10 µm. The region of interest is indicated with a white square and magnified in insets. Scale bars in the higher magnifying images: 2 µm. (**C**,**D**) Quantitative analyses of the numbers of αSyn aggregates and αSyn/FABP5 positive aggregates. Results are presented as means ± SEM (*n* > 40 cells from three respective cell experiments). ** *p* < 0.01, vs. 0.1 μM rotenone-treated αS/F5 cells; ## *p* < 0.01, vs. 0.5 μM rotenone-treated αS/F5 cells.

**Figure 4 biomedicines-09-00110-f004:**
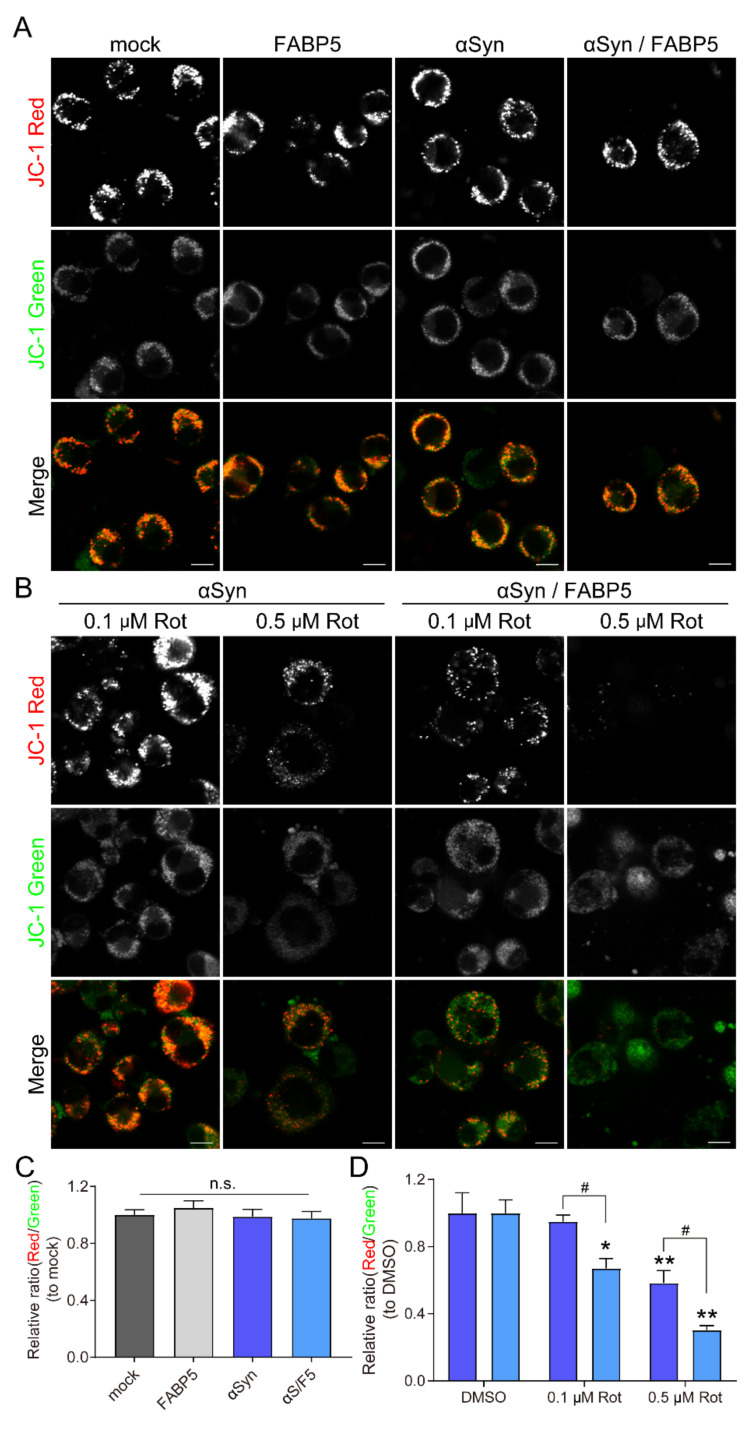
FABP5 involvement in rotenone-induced reduction of mitochondrial activity. (**A**,**B**) Representative images of JC-1-stained Neuro-2A cells. The decrease in the ratio of red to green fluorescence of JC-1 dye was indicated by a reduction in mitochondrial membrane potential. (**A**) Cells were transfected with pcDNA 3.1, *FABP5, αSyn,* and *αS/F5*, respectively, then exposed to DMSO. (**B**) αSyn and αS/F5 transfected cells were exposed to 0.1 or 0.5 μM rotenone. (**C**,**D**) Graphs demonstrating the quantification of the ratio of JC-1 red to green fluorescent intensity in Neuro-2A cells. Results are presented as means ± SEM (n > 300 cells in C and *n* = 5 wells in D from parallel cell experiments). * *p* < 0.05, ** *p* < 0.01, vs. DMSO-treated groups; # *p* < 0.05, vs. *αSyn*-transfected groups. n.s., not statistically significant. Scale bars: 10 µm.

**Figure 5 biomedicines-09-00110-f005:**
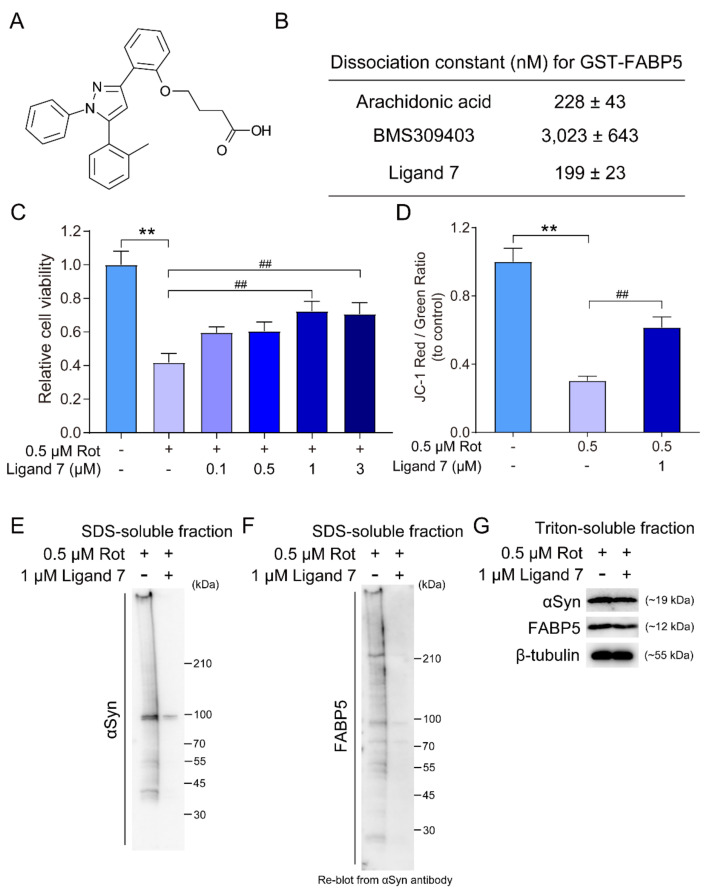
Effects of FABP5 ligand 7 on cell death and αSyn oligomerization and aggregation. All experiments in Figure 5 were carried in *αS/F5*-transfected cells. (**A**) Chemical structure of ligand 7. (**B**) Dissociation constants (*Kd* values, nM) of arachidonic acid, BMS309403, and ligand 7 for GST-FABP5 [24]. (**C**) αS/F5-transfected cells were exposed to 0.5 μM rotenone and the indicated concentrations of ligand 7 for 48 h. Cell viability was measured using CCK-8 assays. Results are presented as means ± SEM (*n* = 6 parallel cell experiments). ** *p* < 0.01, vs. non-rotenone-treated cells; ## *p* < 0.01, vs. rotenone-treated cells without ligand. (**D**) Graphs demonstrating quantification of the JC-1 ratio (red/green). Results are presented as means ± SEM (*n* = 5 parallel cell experiments). ** *p* < 0.01, vs. non-rotenone-treated cells; ## *p* < 0.01, vs. rotenone-treated cells without ligand. (**E**,**F**) Representative images of immunoblots developed with antibodies against αSyn and FABP5 in *αS/F5* cells treated with ligand 7. Three parallel cell experiments were performed with similar results. (**G**) The monomers of αSyn and FABP5 remain in Triton-soluble fractions. Incubation with β-tubulin antibody was used as a protein-loading control.

**Figure 6 biomedicines-09-00110-f006:**
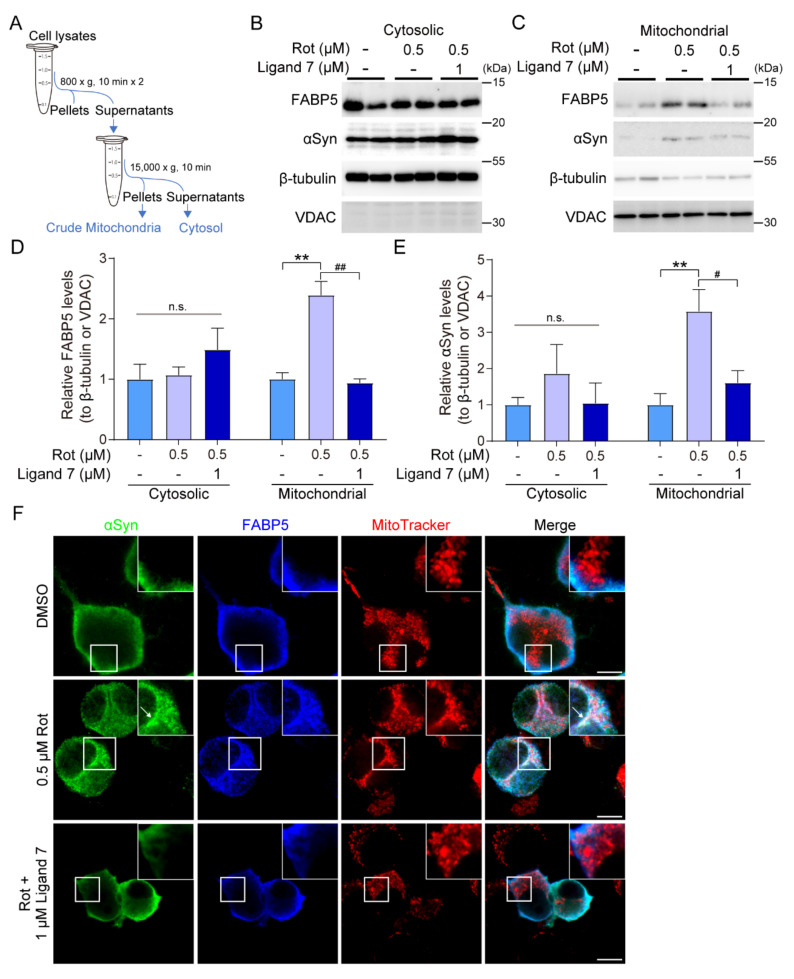
Abnormal accumulation of FABP5 and αSyn in mitochondria was prevented by ligand 7 treatment. All experiments were carried in *αS/F5*-transfected cells. (**A**) Simple diagram of the mitochondrial isolation process in *αS/F5*-transfected cells. (**B**,**C**) Representative images of immunoblots probed with antibodies against FABP5 and αSyn. Incubation with anti-β-tubulin or VDAC antibodies were used as protein-loading controls for cytosolic or mitochondrial fractions, respectively. (**D**,**E**) Quantitative densitometric analyses of αSyn and FABP5 in different fractions. Results are presented as means ± SEM (n = 4, two respective cell experiments, each with 2 parallel dishes). ** *p* < 0.01, vs. non-rotenone-treated cells; # *p* < 0.05 and ## *p* < 0.01, 1 μM vs. rotenone-treated cells without ligand. (**F**) Confocal images of αSyn (green), FABP5 (blue), and MitoTracker (red) in *αS/F5*-transfected cells. The region of interest is indicated with a white square and magnified in insets. Arrow indicates αSyn aggregates which approach and gather in mitochondria with FABP5 in rotenone-treated cells. There were no significant αSyn- and FABP5-positive aggregates in mitochondria of ligand 7-treated cells. Scale bars: 10 µm.

**Figure 7 biomedicines-09-00110-f007:**
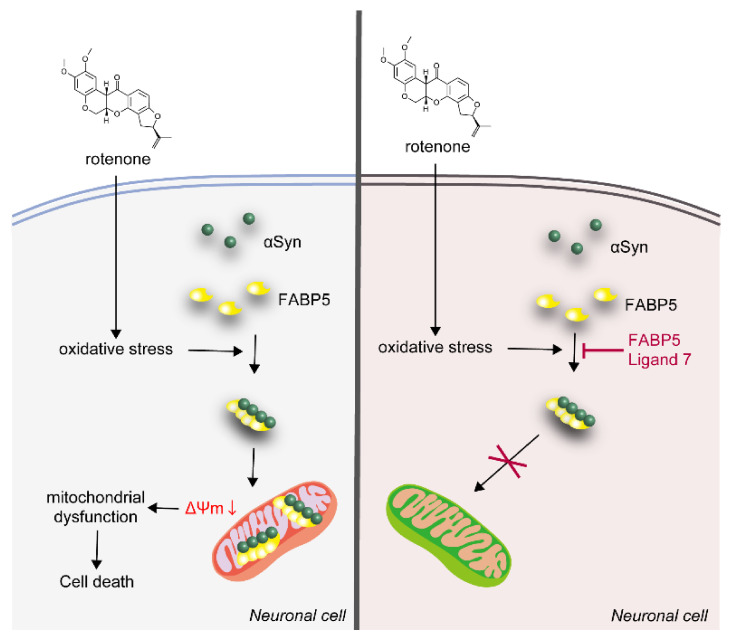
Schematic diagram of the mechanism by which αSyn/FABP5 aggregates aggravate mitochondrial dysfunction and induce neuronal cell apoptosis. FABP5 ligand 7 binds with FABP5 and blocks aggregate formation to attenuate neurotoxicity. (**Left**) when exposed to rotenone, αSyn accumulated with FABP5 to form oligomers and aggregates, which were further targeted to mitochondria. The reduced mitochondrial membrane potential indicated mitochondrial dysfunction which induced cell death. (**Right**) pharmacological inhibition of FABP5 by ligand 7 prevented αSyn/FABP5 aggregate formation, thereby rescuing neuronal cells.

## Data Availability

The data presented in this study are available on request from the corresponding author.

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
