# Peer review of "Epidermal Fatty Acid-Binding Protein 5 (FABP5) Involvement in Alpha-Synuclein-Induced Mitochondrial Injury under Oxidative Stress"

_biomedicines, 2021, doi:10.3390/biomedicines9020110_

Round 1

Reviewer 1 Report

The aim of the present study was to investigate the FABP5 involvement in alpha-syn-induced mitotchondrial injury under oxidative stress. I have the following comments.

  1. The concentration of Rotenone varies between the different tests. Please explain why you chosen the Rotenone concentration 0.1 uM in test; 1. Rotenone induces α-Synuclein and FABP5 oligomerization and aggregation (section starting at line 261), while there is another concentration in e.g. test “3.5. FABP5 ligand rescues rotenone-induced cell death and impedes αSyn oligomerization and aggregation (section starting at line 261)”.
  2. The number of tests performed, i.e what N represents, is not obvious. Do the author refer to number of cell experiments run in parallell, e.g cell experiment is repeated, or how many times each cell experiment was tested with the analytical tool (e.g. WB) or alternative explanation.
  3. Western blotting:

Is there any negative control (unspecific bands) of WB performed, i.e. incubation without primary antibody (alpha-syn or FABP) and subsequently probing with secondary antibody?

How many samples were used for quantification (N=?)? Please provide detailed information regarding the quantification procedure.

See fig 2. Have you investigated the effects on alpha-syn oligomerization and aggregation in alpha-syn transfected cells alone? It would be nice to compare the alpha-syn pattern (WB) between FABP5/alphasyn-transfected Neuro-2A cells, alpha-syn alone and FABP alone.

It would also interesting to similar in figure 5 include both triton and SDS blots of the at 0.5 uM rotenone inhibition.

FABP5/alphasyn-transfected Neuro-2A cells and alpha-syn alone rotenone with WB at 0.5 rotenone concentration and exhit similar WB conditions like figure 5.

Please explain what is mean with (line 123)

The volume of the 2% SDS solution was proportional to the concentration of Triton-soluble protein.

DTT, that breaks possible disulphide linkages, seems only be present during the preparation of the membrane bound cell extract (Triton) and not in the SDS extract.

Please discuss if there are some limitations of your study.

  1. Please check the following sentences in the abstract (line 16), the content might be somewhat re-formulated.

In the presence of rotenone, a mitochondrial respiratory chain complex I inhibitor, 16 co-overexpression of FABP5 with αSyn significantly decreased the viability of Neuro-2A cells compared to that of αSyn alone. Under these conditions, FABP5 co-localized with αSyn in the mitochon-18 dria, consequently reducing mitochondrial membrane potential.

  1. Figure 6 (B and C) Check if the sample labelling is correct.

Author Response

Reviewer 1:

The aim of the present study was to investigate the FABP5 involvement in alpha-syn-induced mitochondrial injury under oxidative stress. I have the following comments.

The concentration of Rotenone varies between the different tests. Please explain why you chosen the Rotenone concentration 0.1 uM in test; 1. Rotenone induces α-Synuclein and FABP5 oligomerization and aggregation (section starting at line 261), while there is another concentration in e.g. test “3.5. FABP5 ligand rescues rotenone-induced cell death and impedes αSyn oligomerization and aggregation (section starting at line 261)”.

Ans: As shown in Figure 1C, under 0.5 μM rotenone treatment, about 50% of the cells have died, so firstly we chosen a moderate concentration at 0.1 μM to analyze protein aggregation under the condition of not being damaged to death. While in the part of 3.5, in order to evaluate the protective effect of ligand, we need an obvious damage condition, so we have chosen 0.5 μM.

The number of tests performed, i.e what N represents, is not obvious. Do the author refer to number of cell experiments run in parallel, e.g cell experiment is repeated, or how many times each cell experiment was tested with the analytical tool (e.g. WB) or alternative explanation.

Ans: N represents number of cell experiments run in parallel in Figure1, Figure 4 and Figure 5; N represents respective cell experimentis repeated in Figure 2; over 40 cells from three respective cell experiments were quantitated in Figure 3; n = 4 represents two respective cell experiments, each with 2 dishes of cells.
And these descriptions have been added to the corresponding legends of figure1 ~ 6.

Western blotting: Is there any negative control (unspecific bands) of WB performed, i.e. incubation without primary antibody (alpha-syn or FABP) and subsequently probing with secondary antibody?

Ans: We added negative control of WB (Figure S2) as shown in supplement file according to the comment. WB membrane was incubated without primary antibody and subsequently probing with secondary antibody, and there is no target band detected.

How many samples were used for quantification (N=?)? Please provide detailed information regarding the quantification procedure.

Ans: In Figure 2, samples from 3 times cell experiments were used for quantification. The description of number of experimental samples has been added in legend of Figure 2. Quantification procedures are as follows: Briefly, we framed each band, also framed the part without bands as the background to get their densities. The difference between the two is the intensity of the target band and is standardized with the internal reference protein (β-tubulin). Finally, the values were normalized with the mock group before statistical analysis.And this detailed information was added into method ‘2.4. Protein extraction and immunoblotting assay’.

See fig 2. Have you investigated the effects on alpha-syn oligomerization and aggregation in alpha-syn transfected cells alone? It would be nice to compare the alpha-syn pattern (WB) between FABP5/alphasyn-transfected Neuro-2A cells, alpha-syn alone and FABP alone. It would also interesting to similar in figure 5 include both triton and SDS blots of the at 0.5 uM rotenone inhibition. FABP5/alpha syn-transfected Neuro-2A cells and alpha-syn alone rotenone with WB at 0.5 rotenone concentration and exhit similar WB conditions like figure 5.

Ans: We didn’t investigate the effects on αSyn oligomerization/aggregation in αSyn transfected cells alone, but mitochondrial membrane potential was lower in αSyn/FABP5-cotransfected cells compared with αSyn-transfected cells (Figure 4B and D) which suggests that FABP5 plays a crucial role in this pathological progression. Furthermore, mitochondrial damage did not cause increased αSyn oligomerization in PC12 cells transfected αSyn alone (Shioda et al., 2014). And we will explore this issue in Neuro-2A cell model in future study.

Please explain what is mean with (line 123). The volume of the 2% SDS solution was proportional to the concentration of Triton-soluble protein.

Ans: During the sample preparation process, we measured the protein concentrations in the supernatants (Triton-soluble protein) obtained after centrifugation. Therefore, when dissolving the remaining pellets, SDS solution was added according to the protein concentration in the supernatant (Triton-soluble protein). So that he total concentration of the SDS-soluble protein obtained in this way is consistent and it can also ensure equivalent protein loading.
We added a short description in the method ‘2.4. Protein extraction and immunoblotting assay’.

DTT, that breaks possible disulphide linkages, seems only be present during the preparation of the membrane bound cell extract (Triton) and not in the SDS extract. Please discuss if there are some limitations of your study.

Ans: Because SDS-soluble fractions were used to evaluate αSyn oligomerization/aggregation, we didn’t add DTT in the SDS extract for a more convincing analysis.

Please check the following sentences in the abstract (line 16), the content might be somewhat re-formulated.

In the presence of rotenone, a mitochondrial respiratory chain complex I inhibitor, 16 co-overexpression of FABP5 with αSyn significantly decreased the viability of Neuro-2A cells compared to that of αSyn alone. Under these conditions, FABP5 co-localized with αSyn in the mitochon-18 dria, consequently reducing mitochondrial membrane potential.

Ans: We rewrote this part as following: In the presence of rotenone, a mitochondrial respiratory chain complex I inhibitor, co-overexpression of FABP5 with αSyn significantly decreased the viability of Neuro-2A cells compared to that of αSyn alone. Under these conditions, FABP5 co-localized with αSyn in the mitochondria, thereby reducing mitochondrial membrane potential.

Figure 6 (B and C) Check if the sample labelling is correct.

Ans: We checked Figure 6 (B and C) and confirmed the sample labelling is correct.

Reviewer 2 Report

The authors clearly demonstrate that FABP5 co-operates with a-Syn to form aggregates under conditions where the mitochondrial respiratory chain is inhibited. They further show that ligand 7 is effective at mopping up FABP5 and thus attenuating the Syn aggregation and subsequent neurotoxicity.

The conclusions are well supported by the data and Fig 7 very nicely summarizes the conclusions of this study. 

Author Response

Reviewer 2:

The authors clearly demonstrate that FABP5 co-operates with a-Syn to form aggregates under conditions where the mitochondrial respiratory chain is inhibited. They further show that ligand 7 is effective at mopping up FABP5 and thus attenuating the Syn aggregation and subsequent neurotoxicity. The conclusions are well supported by the data and Fig 7 very nicely summarizes the conclusions of this study. 

Ans: Thank you for your encouragement.